# Recent Updates on the Synthesis of Bioactive Quinoxaline-Containing Sulfonamides

**Ali Irfan** [1,*] **, Sajjad Ahmad** [2] **, Saddam Hussain** [3] **, Fozia Batool** [4] **, Haseeba Riaz** [4] **, Rehman Zafar** [5] **, Katarzyna Kotwica-Mojzych** [6] **and Mariusz Mojzych** [7,*]

[1] Department of Chemistry, Faculty of Physical Sciences, Government College University Faisalabad, 5-Km, Jhang Road, Faisalabad 38040, Pakistan

[2] Department of Chemistry, UET Lahore, Faisalabad Campus, Faisalabad 37630, Pakistan; sajjad.ahmad@uet.edu.pk

[3] School of Biochemistry, Minhaj University Lahore, Lahore 54000, Pakistan; sad424242@gmail.com

[4] Department of Chemistry, Faculty of Sciences, The University of Lahore, Sargodha Campus, 10 Km, Lahore Road, Sargodha 40100, Pakistan; foziab1996@gmail.com (F.B.); haseebariaz786@gmail.com (H.R.)

[5] Department of Pharmaceutical Chemistry, Faculty of Pharmaceutical Sciences, Riphah International University, Islamabad 44000, Pakistan; rehmanzafar016@gmail.com

[6] Department of Histology, Embryology and Cytophysiology, Medical University of Lublin, Radziwiłłowska 11, 20-080 Lublin, Poland; katarzynakotwicamojzych@umlub.pl

[7] Department of Chemistry, Siedlce University of Natural Sciences and Humanities, 3-go Maja 54, 08-110 Siedlce, Poland

* Correspondence: raialiirfan@gmail.com (A.I.); mariusz.mojzych@uph.edu.pl or mmojzych@yahoo.com (M.M.)

**Abstract:** Quinoxaline is a privileged pharmacophore that has broad-spectrum applications in the fields of medicine, pharmacology and pharmaceutics. Similarly, the sulfonamide moiety is of considerable interest in medicinal chemistry, as it exhibits a wide range of pharmacological activities. Therefore, the therapeutic potential and biomedical applications of quinoxalines have been enhanced by incorporation of the sulfonamide group into their chemical framework. The present review surveyed the literature on the preparation, biological activities and structure-activity relationship (SAR) of quinoxaline sulfonamide derivatives due to their broad range of biomedical activities, such as diuretic, antibacterial, antifungal, neuropharmacological, antileishmanial, anti-inflammatory, anti-tumor and anticancer action. The current biological diagnostic findings in this literature review suggest that quinoxaline-linked sulfonamide hybrids are capable of being established as lead compounds; modifications on quinoxaline sulfonamide derivatives may give rise to advanced therapeutic agents against a wide variety of diseases.

**Keywords:** anticancer; anti-inflammatory; antimicrobial; quinoxaline sulfonamides; synthesis

## 1. Introduction

Heterocyclic scaffolds hold a pivotal place in medicinal chemistry and are one of the largest areas of research in organic chemistry due to their wide spectrum of biological activities. Therefore, continuous efforts have been undertaken to explore their therapeutic potential in the field of design and development of new drugs. Quinoxaline **1** is a privileged, ubiquitous and versatile nitrogen-containing heterocyclic motif, which exhibits a broad spectrum of medicinal, pharmacological and pharmaceutical applications. Quinoxaline, also known as benzopyrazine, was formed by the combination of pyrazine **2** and benzene **3** rings at the carbons 2 and 3 of the pyrazine ring (Figure 1) [1–5]. Quinoxaline derivatives are of relevant interest in medicinal chemistry because of a wide range of pharmacological and biological activities, such as antitumor, anticancer [6–12], antidiabetic [13–17], antimycobacterium tuberculosis [18,19], antimicrobial [20–22], antidepressant [23], anthelmintic [24], analgesic [25–31], anti-inflammatory [11,32–36], antiviral [37–43], antifungal [44–49], antimalarial [50–55], antibacterial [56–63], antioxidant [14,64,65], antithrombotic [66,67] and antiprotozoal [54,68].

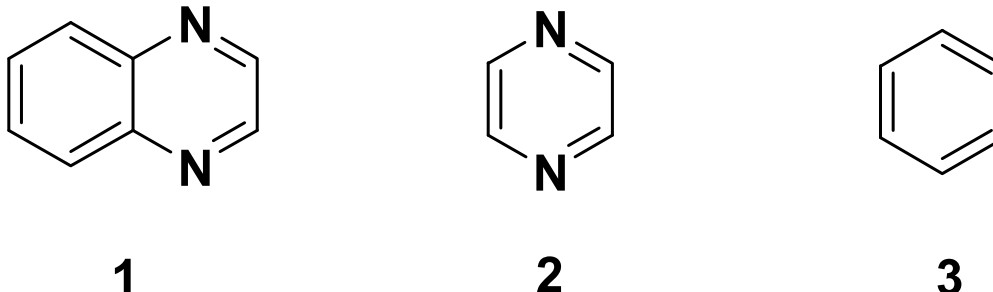

**Figure 1.** Structures of quinoxaline, pyrazine and benzene.

Different quinoxaline-based drugs have been playing a main role in the therapy and treatment of various diseases, such as varenicline **4** (aids in smoking cessation), brimonidine **5** (anti-glaucoma activity), quinacillin **6** (antibacterial activity) and XK469 NSC **7** (selective topoisomerase IIβ poison) (Figure 2) [69–71].

**Figure 2.** Structures of quinoxaline drugs **4–7**.

Sulfonamides were first characterized in 1932 and are organic chemical frameworks in which a sulfonyl group is linked to an amine group (-$SO_2$-$NR^1R^2$) [72]. The sulfonamide group is a key moiety that plays a paramount role in both medicinal chemistry and organic chemistry due to their immense biological action. A series of therapeutic agents contain a sulfonamide moiety in their chemical structures [73–79]. Particular examples are antibiotics which are used on a large scale for many clinical purposes [80]. Heterocyclic sulfonamides exhibit different biological activities, such as antibacterial [81–83], anti-inflammatory [84], antibacterial [85], antimicrobial [86], antifungal [87], antiprotozoal [88], antiviral [89], antimalarial [90], antitumor [91–93], carbonic anhydrase inhibitors [94–99], antidiabetic [100], anticonvulsant, [101], anti-glaucoma [95,97], anti-obesity [102], diuretic [103–105], hypoglycemic [95,97], anti-neuropathic pain [106], matrix metalloproteinase and bacterial protease inhibitors [107,108]. The sulfonamide moiety is a basic component of drugs such as sulfamethazine 8 (antibacterial), chlortalidone 9 (diuretic), sulfametopyrazine 10 (chronic bronchitis, urinary tract infections and malaria), chlorpropamide 11 (treats diabetes mellitus type 2), ethoxzolamide 12 (carbonic anhydrase inhibitor) and mafenide 13 (antimicrobial agent used to treat severe burns) (Figure 3) [109].

Hybrid structures formulated by the combination of quinoxaline and sulfonamide moieties display novelty and versatility and possess a consistent therapeutic potential against most diseases. The general structure of quinoxaline sulfonamide 14, sulfaquinxaline 15 (antimicrobial and a coccidiosis for veterinary use) and chloroquinoxaline sulfonamide 16 (topoisomerase-IIα and a topoisomerase-IIβ poison) are depicted in Figure 4 [110].

The significance of quinoxline sulfonamide derivatives in medicinal chemistry as therapeutic agents is clearly displayed by the patents. The patented sulfonamide derivatives, such as substituted quinoxaline-2,3-diones 17 (glutamate receptor antagonists) [111], quinoxaline-containing pyridine-3-sulfonamide derivative 18 (used as PI3K inhibitors) [112], amidophenyl-sulfonylamino-quinoxaline compounds 19 (CCK2 modulators useful in the treatment of CCK2 mediated diseases) [113], quinoxaline compounds **20** (for the treat-

ment of autoimmune disorders and/or inflammatory diseases and cardiovascular diseases, etc.) [114], dichlorophenyl moiety-containing quinoxaline sulfonamide **21** (useful for the treatment of disease states mediated by CCK2 receptor activity) [115] and benzimidazole moiety-based quinoxaline benzene sulfonamide scaffold **22** (phosphatidylinositol 3-kinase inhibitors) [116], 2-chloro-5-methoxyphenylamino based quinoxaline sulfonamide derivative **23** (phosphatidylinositol 3-kinase inhibitors) [117], substituted quinoxaline compound **24** (HCV NS3 protease inhibitors) [118], macrocyclic quinoxaline compound **25** (HCV NS3 protease inhibitors) [119], benzene sulfonamide derivatives of quinoxaline **26** (anticancer agents) [120], tetrahydro quinoxaline derivative **27** (inhibitors of sodium channels/pain disorders) [121], benzene sulfonamide derivatives of quinoxaline **28** (kinase inhibitors, anticancer agents) [122], quinoxaline sulfonamides **29** (PI3K inhibitors, anti-inflammatory and autoimmune diseases) [123], substituted aminoquinoxaline sulfonamides **30** and substituted chloroquinoxaline sulfonamides **31** [124], substituted 1,2,4-thiadiazole based quinoxaline sulfonamide derivative **32** (inhibitors of sodium channels) [125] and substituted diphenyl quinoxaline sulfonamide derivative **33** (inhibitors of NAMPT) [126] are depicted in Figure 5.

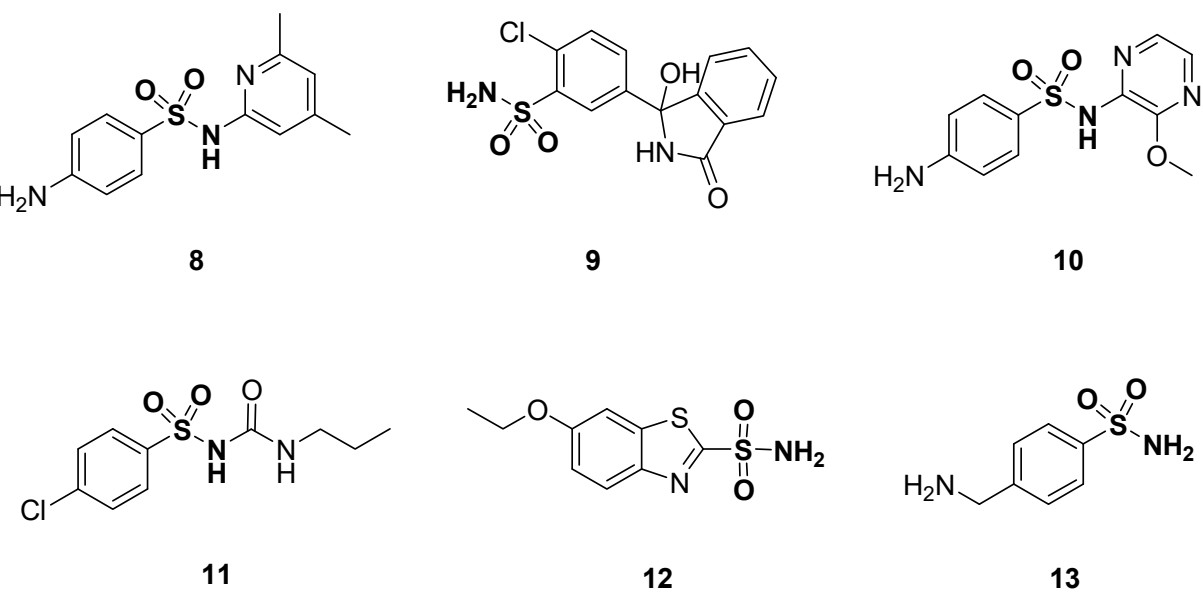

**8** **9** **10**

**11** **12** **13**

**Figure 3.** Structures of sulfonamide drugs **8**–**13**.

**14** **15** **16**

**Figure 4.** Structures of quinoxaline sulfonamide drugs **14**–**16**.

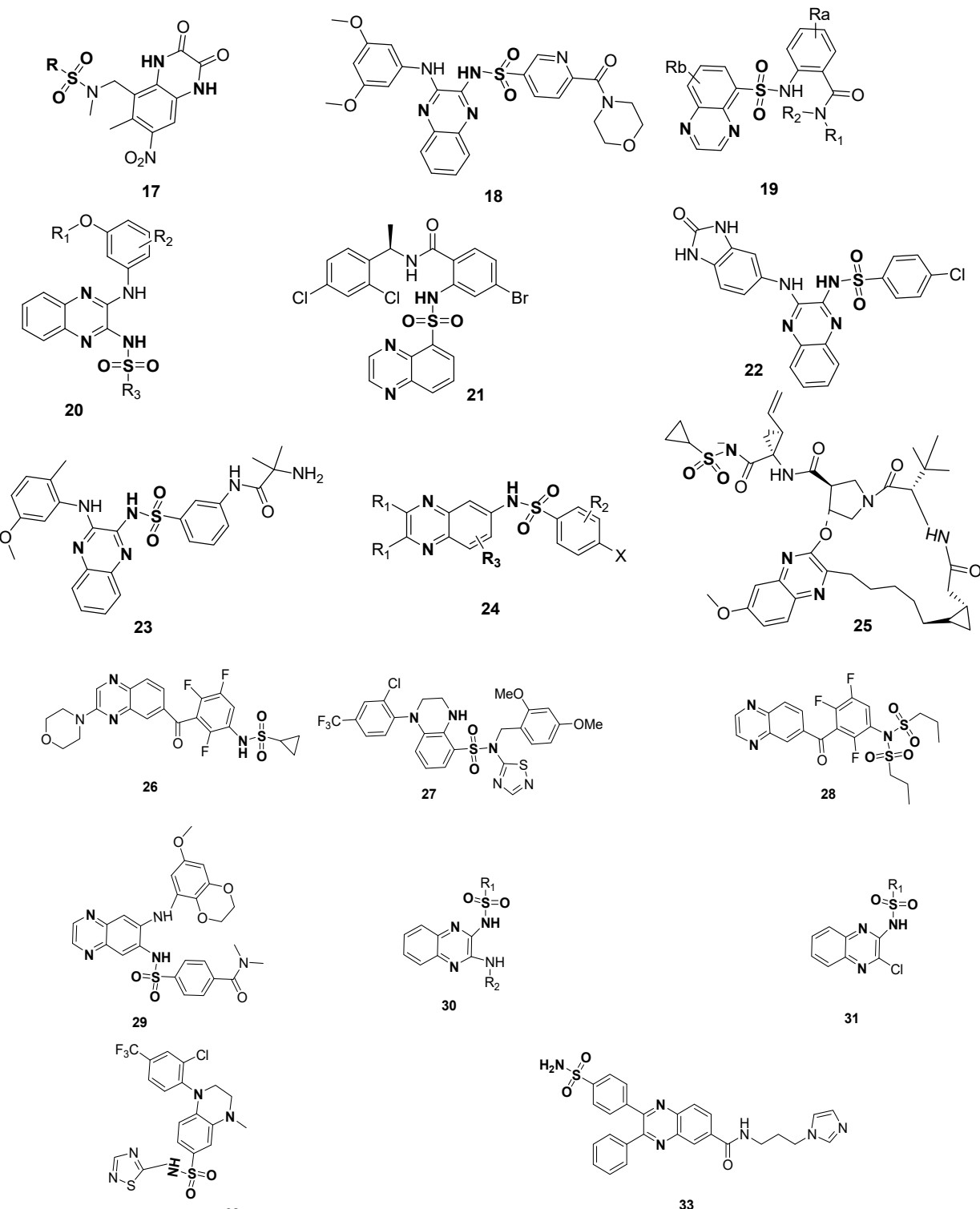

**Figure 5.** Structures of patented quinoxaline sulfonamides **17–33**.

## 2. Synthesis and Biological Activities

Standard reference works have shown that the *orthro*-phenylenediamine (OPD) can be condensed with dicarboxylic acids, diketones, α-halo-ketones and esters to give quinoxalines. The development of quinoxaline sulfonamide chemistry is linked with the presence of amino (-NH$_2$) and sulfonyl chloride (-SO$_2$Cl) groups in the reacting species. Commonly, quinoxaline sulfonamides can be available via general transformation of substituted amines,

with the quinoxaline containing sulfonyl chloride functionality and vice versa, as depicted in Figure 6.

**Figure 6.** General preparation of quinoxaline sulfonamides.

### 2.1. Benzothiazole Quinoxaline Sulfonamide Derivatives with Diuretic Activity

Medications designed to increase the excretion of water and salt in urine are termed as diuretics. Diuretics are used as therapeutics in liver cirrhosis, epilepsy, edema, hypertensions, heart failure, hypercalciuria, diabetes insipidus and in some kidney diseases. Their mechanism of action on distinct sites of nephrons involves the production of diuresis and inhibition of sodium ions re-absorption in the renal tubules of the kidney.

A new multistep synthetic methodology was developed for the synthesis of a novel promising class of substituted quinoxaline sulfonamides as diuretic agents, assessed in vivo by Husain and coworkers [127]. The condensation of *o*-phenylene diamine **34** with pyruvic acid **35** in methanol produced 3-methylquinoxaline-2*H*-one **36** with an 80% yield that was further reacted with substituted aldehydes **37** in the presence of piperidine to afford the substituted quinoxaline derivatives **38**. The condensation of 2-amino-benzothiazole-6-sulfonamide (**41**) with **38** in ethanol yielded the final quinoxaline sulfonamide derivatives **42** in moderate to good yield (60–82%) (Scheme 1) [127]. The derivatives with electron-donating groups showed the maximum yield, while the electron-withdrawing groups resulted in a minimum yield.

The Lipschitz method was adopted to determine the diuretic activity of the thiazole moiety-based quinoxaline sulfonamide hybrids. Fifty-four healthy adult Wistar albino rats (180–200 g) were selected and divided into nine groups, and these were acclimatized in standard environmental conditions for one week.

The thiazole moiety-containing quinoxaline sulfonamide derivative **43** showed a remarkably high diuretic activity and Lipschitz value (1.13 and 1.28 values, respectively) compared to the standard reference drugs, acetazolamide and urea. The derivative **44** exhibited a moderate diuretic activity (value of 0.78) and Lipstchitz value (0.88). The structural motif **45** exhibited the least diuretic action among the three compounds (value of 0.56) and a Lipschitz value of 0.63 (Table 1). These results showed that diuretic activity is directly proportional to the Lipschitz value. The SAR showed that the presence of a strong electron-donating group (EDG), such as methoxy, on the benzene ring of scaffold **43** increases the diuretic activity, while the presence of strong electron-withdrawing group (EWG), such as an -$NO_2$ group, decreases the diuretic activity of the analog **45**. The presence of an -$N(CH_3)_2$ moiety on the benzene ring of derivative **44** induced a moderate diuretic effect (Figure 7). The SAR study revealed that the diuretic activity decreased in order of –$OCH_3$ > -$N(CH_3)_2$ > -$NO_2$. This indicated that EDG increases the diuretic activity, while EWG decreases the diuretic activity [127].

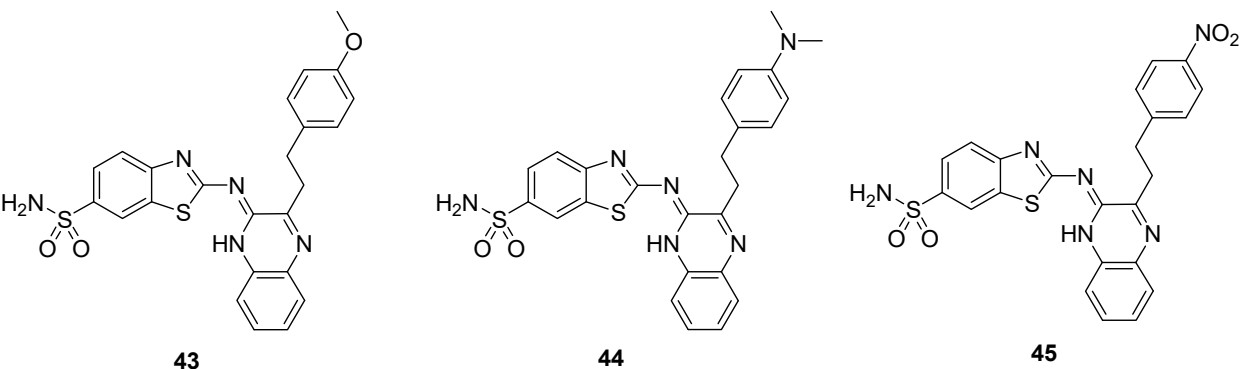

**Scheme 1.** Synthesis of sulfonamide derivatives **42**.

**Table 1.** Diuretic activity of benzothiazole quinoxaline sulfonamide derivatives **43**–**45**.

| Compound | Urinary Excretion Percentage | Diuretic Activity | Lipschitz Value |
|---|---|---|---|
| 43 | 176.66 | 1.13 | 1.28 |
| 44 | 121.42 | 0.78 | 0.88 |
| 45 | 87.5 | 0.56 | 0.63 |
| Acetazolamide | 155.17 | 1.0 | 1.13 |
| Urea | 136.6 | 0.88 | 1.0 |

**Figure 7.** Substituted quinoxaline benzothiazole sulfonamide derivative **43**–**45**.

## 2.2. Quinoxaline Sulfonamides with Antibacterial Activity

Alavi et al. reported a facile, efficient, solvent and catalyst-free green protocol for the synthesis of quinoxaline sulfonamide derivatives, which were screened for their antibacterial activity against different Gram-positive and Gram-negative bacterial strains. The quinoxaline sulfonyl chloride (QSC) **48** was synthesized in 85% yields by the treatment of methoxyphenyl quinoxaline **46** with chlorosulfonic acid **47**. The QSC scaffold **48** was reacted with substituted aromatic amines in neat and ecofriendly conditions to afford substituted quinoxaline sulfonamides of the type **49** in good to excellent yield (Scheme 2).

Aromatic amines with EDG, such as methyl and methoxy, reacted in 3–10 min and lead to products with a higher yield, while aromatic amines with EWG afforded products with a low yield. Aromatic amines with strongly EWG, such as nitro, do not react in the below mentioned conditions [128].

Green effeicient and solvent free approach to afford derivatives in good to excellent 74-92% yield

**R** = H, C$_2$H$_5$; **R$_1$** = Aryl

**Scheme 2.** Synthesis of substituted quinoxaline sulfonamide derivatives, with structure **49**.

The substituted 4-methoxyphenyl quinoxaline sulfonamide **50** and substituted 2-methoxyphenyl quinoxaline sulfonamide derivative **51** showed the best antibacterial activity against *S. Aureus*, having a zone of inhibition (ZOI) of 20 mm and 22 mm, respectively, which is more than the standard chloramphenicol drug (ZOI 19 mm) but less than reference ampicillin drug (ZOI 28 mm). The substituted naphthalene quinoxaline sulfonamide derivatives **52** exhibited the highest antibacterial activity against *E. coli*, with a ZOI value of 18 mm when compared to the standard drug, ampicillin (ZOI 15 mm), but less than the reference chloramphenicol drug (ZOI 19 mm) (Table 2). A moderate antibacterial activity was observed for the substituted chlorophenyl quinoxaline sulfonamides **53** and **54** against Gram-positive *S. Aureus* and Gram-negative *E. coli.* The unsubstituted phenyl quinoxaline sulfonamide derivative **55** showed the least antibacterial activity against *S. Aureus* and *E. coli*, exhibiting a ZOI of 5 mm and 5> mm, respectively, when compared to the reference drugs, chloramphenicol and ampicillin, exhibiting ZOI values of 19 mm and 28 mm against *S. Aureus* and 22 mm and 15 mm against *E. coli*. The SAR revealed that the presence of electron-rich groups on the aminophenyl ring showed the highest antibacterial activity for derivatives **50–52**, whereas the presence of the electron-withdrawing groups on the aminophenyl ring, such as a chloro atom (weak EWD group), in **53** and **54** scaffolds led to a moderate antibacterial activity, while the least antibacterial activity was observed for the unsubstituted derivative **55** (Figure 8) [128]. The results showed that electron-donating group's substitution on the quinoxaline sulfonamide scaffolds contribute positively to increase the antibacterial activity.

**Table 2.** Antibacterial activity of quinoxaline benzene sulfonamide derivatives **50–55**.

| | Antibacterial Activity (DMSO, 4 mg mL$^{-1}$) | |
|---|---|---|
| | Diameter of Zone of Inhibition (mm) | |
| Compound | *S. Aureus* | *E. coli* |
| 50 | 20 | 9 |
| 51 | 22 | 11 |
| 52 | 17 | 18 |
| 53 | 15 | 12 |
| 54 | 11 | 13 |
| 55 | 5 | 5> |
| Chloramphenicol | 19 | 22 |

**Figure 8.** Substituted quinoxaline benzene sulfonamide derivatives **50–55**.

Ingle et al. synthesized antimicrobial quinoxaline sulfonamide derivatives by applying a convenient and an expeditious methodology. In this synthetic strategy, the nucleophilic substitution was performed by attacking the electrophilic center (S) of chloro sulfonic acid **47** by the benzene of diphenyl quinoxaline **56** to furnish the intermediate 2,3-diphenylquinoxaline-6-sulfonyl chloride **57**, which further refluxed with the substituted primary amine under basic conditions to deliver the final product diphenyl quinoxaline sulfonamide **58** in moderate to good (69–83%) yield (Scheme 3). The aromatic amines with EWG, such as nitro, lead to the final products in a higher yield, while the aromatic amines with EDG, such as methyl and methoxy, etc, lead to final products in low yield [129].

The quinoxaline sulfonamide analogs **59** displayed the highest antibacterial activity, exhibiting a ZOI value of 10 mm against *S. aureus* and 8 mm against *E. coli* at a concentration of 100 μg (microgram), while the quinoxaline sulfonamide hybrid **60** showed moderate antibacterial activity, with a ZOI of 7 mm and 6 mm at a concentration of 100 μg against *S. aureus* and *E. coli*, respectively (Table 3). The moderate antibacterial activity was observed for derivative **61** that exhibited a ZOI value of 6 mm against *S. aureus* at a concentration of 100 μg, and derivative **62** exhibited the least antibacterial activity by inducing a 2 mm ZOI value at the concentration of 100 μg against *E. coli*, when compared with the standard drug azithromycin (ZOI value of 12 mm and 10 mm against *S. aureus* and *E. coli* at the concentration of 100 μg). The SAR revealed that the maximum antibacterial activity was observed in the presence of a strong EWD group (chloro atom) on the ring of analogue **59**, while the introduction of a *m*-OH and *p*-OH group on the ring produced a moderate activity for compounds **60** and **61**. The derivative **62** exhibited the least antibacterial activity due to the presence of an electron-denoting methoxy group at the phenyl ring (Figure 9). The presence of EWG showed the highest antibacterial activity as compared to the presence of EDG, which displayed moderate to low antibacterial activity [129].

**Expeditious and convenient synthetic approch to afford products in moderate to good (69–83%) yield**

**Scheme 3.** Synthesis of substituted quinoxaline sulfonamide derivatives with structure **58**.

**Table 3.** Antibacterial activity of quinoxaline-based substituted benzene sulfonamide derivatives.

| Compound | Zone of Inhibition (mm) at 100 µg | |
|---|---|---|
| | *S. aureus* | *E. coli* |
| 59 | 10 | 8 |
| 60 | 7 | 6 |
| 61 | 6 | 7 |
| 62 | 9 | 2 |
| Azithromycin | 12 | 10 |

**Figure 9.** Substituted quinoxaline sulfonamide derivatives **59–62**.

Talari and coworkers developed a novel synthetic strategy for the synthesis of indole-based quinoxaline sulfonamides. These sulfonamide derivatives were evaluated for their antimicrobial activity against different bacterial and fungal strains. Indole quinoxaline ana-

logue **64** was afforded by the reaction of OPD **34** with 5-bromoisatin **63** heated to reflux for 30 min. The scaffold **64** was refluxed with chlorosulfonic acid **47** to obtain the quinoxaline sulfonyl chloride **65** that was further treated with substituted amines in anhydrous acetone and pyridine to furnish final quinoxaline sulfonamides **66** via dehydrohalogenation in a low to moderate (20–47.5%) yield (Scheme 4) [130].

**Scheme 4.** Synthesis of substituted quinoxaline sulfonamide derivatives with structure **66**.

Some of the synthesized quinoxaline sulfonamide derivatives were screened for antimicrobial activities against some bacterial and fungal strains. The quinoxaline sulfonamide **67** showed the most potent antibacterial and antifungal activity, with ZOI values of 14 mm, 14 mm, 16 mm, 17 mm and 17 mm against *S. aureus*, *B. pimilis*, *E. coli A. niger* and *P. notatum*, respectively at 50 μg/mL (Table 4). The quinoxaline sulfonamide derivative **68** demonstrated a moderate activity, exhibiting ZOI values of 13 mm against *S. aureus*, 14 mm against *B. pimilis*, 15 mm against *E. coli* and 12 mm against *A. niger* at 50 μg/mL. The derivative **69** exhibited the lowest ZOI values of 10 mm against *B. pimilis* and 10 mm against *A. niger* at 50 μg/mL. The reference standard streptomycin at 50 μg/mL exhibited a ZOI of 18 mm, 20 mm and 20 mm against *S. aureus*, *B. pimilis* and *E. coli* strains, respectively. Miconazole nitrate exhibited, instead, 23 mm and 20 mm ZOI values at 50 μg/mL against *A. niger* and *P. notatum* strains. The SAR showed that the introduction of a bromophenyl moiety on the sulfonamide increased the antimicrobial activity of **67**, while thioamide functionality was incorporated in analogue **68** to induce a moderate antimicrobial activity. The acetophenone moiety-containing derivative **69** displayed the least antimicrobial activity (Figure 10). The SAR study indicated that the antimicrobial activity decreased in the order Bromo > Thiamide > Acetyl. The results in the Table 4 showed that the structural motif **67** displayed a broad spectrum antimicrobial therapeutic potential against different bacterial and fungal strains than the reference standard drugs [130].

**Table 4.** Antimicrobial activity of quinoxaline-substituted benzene sulfonamide derivatives **67**–**69**.

| | Zone of Inhibition (mm) | | | | |
|---|---|---|---|---|---|
| **Compound** | **Antibacterial** | | | **Antifungal** | |
| | *S. aureus* | *B. pimilis* | *E.coli* | *A. nigar* | *P. notatum* |
| 67 | 14 | 14 | 16 | 17 | 17 |
| 68 | 13 | 14 | 15 | 12 | - |
| 69 | - | 11 | - | 10 | - |
| Streptomycin | 18 | 20 | 20 | - | - |
| Miconazole nitrate | - | - | - | 23 | 20 |

**Figure 10.** Substituted quinaxoline sulfonamide derivatives **67–69**.

An efficient pathway was adopted by Sharaf El-Din et al. to afford substituted quinoxaline sulfonamide derivatives, and they analyzed these designed scaffolds for antibacterial therapeutic potential by the agar diffusion method. The 1,4-dihydroquinoxaline-2,3-dione (**71**) was prepared by condensation of **34** with oxalic acid **70**. In the next step, reaction with chlorosulfonic acid (**47**) at 0–5 °C furnished the quinoxaline scaffold with chlorosulfonyl moiety **72**, which was combined with different amino compounds, such as amines, amino acids, morpholine and piperazine, to produce final quinoxaline sulfonamide derivatives **73** in good yield (61–66%) (Scheme 5) [131].

R = NHC$_3$H$_7$, NHC$_3$H$_6$OH, NHCH$_2$CO$_2$H, NH$^-$(COOH-C$_6$H$_4$), morpholine, piperazine

**Scheme 5.** Synthesis of quinoxaline-2,3 (1*H*,4*H*) dione sulfonamide derivative with structure **73**.

The quinoxaline sulfonamide **74** displayed the highest antibacterial activity, with a zone of inhibition (ZOI) of 15 mm and 10 mm against *S. aureus* and *E. coli*, respectively, while the reference drug, sulfadiazine, gave a ZOI of 14 mm and 13 mm zone diameter against *S. aureus* and *E. coli* (Table 5). The derivative **75** showed equipotent antibacterial activity (ZOI of 14 mm) against *S. aureus* in comparison with the reference drug (ZOI of 14 mm), but less antibacterial activity (ZOI of 8 mm) against *E. coli* than the reference drug (ZOI of 13 mm). A moderate activity was observed for the quinoxaline sulfonamide derivative **76** that led to a ZOI of 12 mm and 10 mm against *S. aureus* and *E. coli*, respectively. Compound **77** exhibited the lowest antibacterial activity, showing a ZOI of 10 mm against *S. aureus* with respect to the reference drug sulfadiazine, but it did not exhibit any activity against *E. coli*. The SAR studies demonstrated that the presence of a propanol or benzoic acid moiety on the sulfonamide analogs **74** and **75** displayed an excellent and slightly higher or equipotent antibacterial activity in comparison with the reference drug sulfadiazine,

while the presence of a sulfonyl glycine moiety was responsible for the moderate activity of the derivative **76**. The morpholine moiety in analog **77** was responsible for the least antibacterial activity against Gram-positive and negative bacterial strains. (Figure 11) [131].

**Table 5.** Antibacterial activity of quinaxoline sulfonamide derivatives **74–77**.

| | Antibacterial Activity (20 mg/mL) | |
|---|---|---|
| **Compound** | **Zone Diameter (mm)** | |
| | *S. aureus* | *E. coli* |
| 74 | 15 | 10 |
| 75 | 14 | 8 |
| 76 | 12 | 10 |
| 77 | 10 | - |
| Sulfadiazine | 14 | 13 |

**Figure 11.** Quinaxoline sulfonamide derivatives **74–77**.

Potey et al. described a new synthetic approach for the synthesis of libraries of quinoxaline sulfonamides and studied their antimicrobial activities. The starting 2,3-diphenylquinoxaline (**56**) was prepared in an excellent yield by condensation of *o*-phenylenediamine (OPD) (**34**) and diketone **78**. Then, quinoxaline **56** was treated with chlorosulfonic acid **47** at room temperature to afford the quinoxaline sulfonyl chloride **57** that reacted with primary and secondary amines to furnish quinoxaline sulfonamides **79** and **80** in good to excellent yield (Scheme 6) [132].

The prepared derivatives were tested against some Gram-positive and negative bacterial strains (Table 6). Among all these derivatives, the quinoxaline sulfonamide **81** exhibited an excellent antibacterial activity, with a ZOI of 30 mm and 24 mm against *P. vulgaries* and *Enterobacteria*, respectively. A moderate activity was observed for **82**, with a ZOI value of 18 mm against *S. aureus*, 18 mm against *Enterobacteria*, 16 mm against *V. cholorie*, 16 mm against *E. coli* and 16 mm against *P. vulgaries.* Compound **83** showed the least antibacterial activity, exhibiting a ZOI of 8 mm, 8 mm, 6 mm, 6 mm and 8 mm against *S. aureus, Enterobacteria*, *V. cholera, E. coli* and *P. vulgaries*, respectively. The SAR showed that *ortho*-OH on the phenylsulfonamide moiety increased the antibacterial activity of the quinoxaline sulfonamide **81**, while the presence of a 2-chlorophenyl substituent was responsible for the moderate activity of the scaffold **82.** The introduction of a methoxyphenyl group into the quinoxaline sulfonamide derivative **83** decreased the antibacterial activity (Figure 12) [132].

**Scheme 6.** Synthesis of 2,3-diphenylquinoxaline sulfonamide derivatives **79** and **80**.

**Table 6.** Antimicrobial activity of quinoxaline derivatives **81–83**.

| | Antimicrobial Activity | | | | |
|---|---|---|---|---|---|
| | Zone of Inhibition (mm) | | | | |
| Compound | Gram-Positive Bacteria | Gram-Negative Bacteria | | | |
| | *S. aureus* | *Enterobacteria* | *V. cholera* | *E. coli* | *P. vulgaris* |
| 81 | 19 | 24 | 2 | 22 | 30 |
| 82 | 18 | 18 | 16 | 16 | 16 |
| 83 | 8 | 8 | 6 | 6 | 8 |

**Figure 12.** Substituted quinaxoline sulfonamide derivatives **81–83**.

### 2.3. Synthesis of Quinoxaline Sulfonamide Derivative with Neuropharmacological Activity

The activity which determines how drugs or therapeutic agents affect the cellular functions in the nervous system is called neuropharmacological activity. The neuropharmacological effects, such as analgesia, sedation, convulsion, anxiety, memory and psychosis, and the neural mechanisms through which the drugs influence behavior were studied in neuropharmacology.

Olayiwola et al. reported the green microwave synthetic route to quinonoxaline sulfonamide with neuropharmacological activity. The 2,3-quinoxalinedione **71** was furnished in an excellent yield (99%) by the condensation of **34** and oxalic acid dihydrate (**70**) using microwave radiations. The scaffold **71** was treated with **47** to obtain quinoxaline-6-sulfonyl chloride **72** at 88% yield, followed by the reaction with dibenzyloamine in anhydrous dimethylformamide (DMF) to give corresponding quinoxaline sulfonamide **84** at 75% yield (Scheme 7) [133].

**Scheme 7.** Synthesis of quinoxaline sulfonamide **84**.

The obtained derivative **84** was analyzed for its neuropharmacological effects, such as anxiolytic, anticonvulsant and total locomotor activity, in mice (Table 7). It showed maximum inhibition of locomotor activity (sedative action) at 40 mg/kg. As for anxiolytic activity, **84** exhibited a maximum action at 2.5 mg/kg, having 67.3% time in open arm and 81.0% entry into open arm, and the index of open arm avoidance was 25.9 when compared with diazepam at 1 mg/kg (53.3% = time in open arm, 82.4% = entry into open arm and 32.2 = index of open arm avoidance). As for anticonvulsant activity, **84** displayed a protection effect of 100% against both leptazol at 80 mg/kg and strychnine at 2.0 mg/kg after 60 min at 25 mg/kg dose, when compared with phenobarbitone sodium, having a 90% protection effect at 20 mg/kg dose against both leptazol at 80 mg/kg and strychnine at 2.0 mg/kg after 60 min [133].

**Table 7.** Neuropharmacological activity of quinoxaline-based substituted benzene sulfonamide derivatives.

| Compound | Activity | Inhibition Effect | Reference Compound | Inhibition Effect |
|---|---|---|---|---|
| | | **Neuropharmacological Effect** | | |
| 84 | Total locomotor activity | Potent sedative at 40 mg/kg | - | - |
| | Anxiolytic activity | 67.3% = time in open arm | Diazepam | 53.3% = time in open arm |
| | | 81.0% = entry into open arm, 25.9 = index of open arm avoidance | | 82.4% = entry into open arm, 32.2 = index of open arm avoidance |
| | | Dose = 2.5 mg/kg | | Dose = 1 mg/kg |
| | Anticonvulsant activity | Protection effect = 100% against Leptazol at 80 mg/kg and Strychnine at 2.0 mg/kg after 60 min | Phenobarbitone sodium | Protection effect = 90% against both Leptazol at 80 mg/kg. and Strychnine at 2.0 mg/kg after 60 min 15.15 |
| | | Dose = 25 mg/kg dose | | Dose = 20 mg/kg |

### 2.4. Quinoxaline Sulfonamide Derivatives with Antileishmanial Activity

Barea et al. synthesized quinoxaline sulfonamide derivatives and studied their antileishmanial activities. The 3-amino-1,4-di-*N*-oxide quinoxaline-2-carbonitrile derivative **89** was synthesized in 15–90% yield by the reaction of benzofuroxane **85** and malononitrile **86**, using triethylamine as catalyst and DMF as solvent. The scaffold **87** was further treated with substituted sulfonyl chlorides **88** at 0 °C to afford quinoxaline sulfonamide derivatives **89** in low yield (13–15%) (Scheme 8) [134].

**Scheme 8.** Synthesis of substituted quinoxaline sulfonamide derivatives **89**.

The synthesized derivatives were evaluated against *Leishmania amazonensis* strain MHOM/BR/76/LTB in infected macrophages (Table 8) The quinoxaline sulfonamide **92** displayed a potent antileishmanial activity (IC$_{50}$ = 3.1 μM), although 15-fold less active than the standard drug amphotericin B. The quinoxaline sulfonamide analog **91** displayed a moderate antileishmanial activity (IC$_{50}$ = 16.3 μM), comparable to that observed for 90 (IC$_{50}$ = 20.3 μM). The SAR demonstrated that the presence of a 2-naphtyl moiety on the sulfonamide motif and a methyl group on the quinoxaline in 90 led to a potent antileishmanial activity, while a 2-naphtyl moiety and the electronegative chlorine atom was responsible for the moderate antileishmanial activity of scaffold 91. The highest antileishmanial activity was observed for **92** with the *o*-nitrophenyl group and chlorine atom (Figure 13) [134].

**Table 8.** Antileishmanial activity of quinoxaline-1,4-dioxide sulfonamide derivatives **90–92**.

| Antileishmanial Activity | |
|---|---|
| Compound | IC$_{50}$ (μM) |
| 90 | 20.3 |
| 91 | 16.3 |
| 92 | 3.1 |
| Amphotericin B | 0.2 |

**Figure 13.** Structure of quinoxaline-1,4-dioxide sulfonamide derivatives **90–92**.

*2.5. Synthesis of Quinoxaline Moiety-Based Benzene Sulfonamides with Antitumor Activity*

A mass or lump of tissue that is formed by an accumulation of abnormal cells, resembling swelling and varying from a tiny nodule to a large mass, is specified as a tumor. Not all tumors are cancerous; some are benign (non-cancerous), premalignant (having potential to become cancerous) and some are malignant (cancerous). A chemotherapeutic agent which has the ability to prevent or inhibit the formation or growth of tumors is termed as antitumor.

Farrag et al. developed a novel synthetic strategy to furnish novel sulfamoyl phenyl carbamoyl quinoxaline hybrid structures. In the first step, quinoxaline-2,3-dicarboxylic

acid **94** was produced by the treatment of OPD 34 with disodium dihydroxytartarate **93**, followed by a reaction with acetic anhydride, which resulted in the formation of quinoxaline anhydride derivative **95**. The scaffold 95 was refluxed with substituted sulfonamide **96** in ethanol to afford 3-sulfamoyl phenylcarbamoyl quinoxaline-2-carboxylic acids **97** at 71–78% yield, which further reacted in quinoline-containing copper powder to obtain corresponding quinoxaline sulfonamide scaffold **98**. The quinoxaline sulfonamide derivative **98** was also prepared at 68–73% yield by directly heating the anhydride compound **95** with sulfonamide compound **96** under fusing conditions (Scheme 9) [135].

**Scheme 9.** Synthesis of quinoxaline moiety-based benzene sulfonamides with structure **98**.

The quinoxaline sulfonamide derivative **99** displayed the most potent antitumor activity against the liver carcinoma cell line (IC$_{50}$ value 0.5 µg m/L), while moderate anti-tumor activity was exerted by **100** (IC$_{50}$ value 4.75 µg m/L, Table 9). The quinoxaline sulfonamide structural motif **101** demonstrated the least antitumor activity, exhibiting an IC$_{50}$ value of 6.79 µg m/L. The SAR described that the presence of a carboxylic acid group on the quinoxaline was responsible for the potent antitumor activity of **99**, while its absence in **100** led to a moderate activity. The absence of the isoxazole in **101** produced the worst action herein (Figure 14) [135].

**Table 9.** Anti-tumor activity of quinoxaline sulfonamides **99–101**.

| Antitumor Activity | |
| --- | --- |
| Compound | IC$_{50}$ (μg m/L) |
| 99 | 0.5 |
| 100 | 4.75 |
| 101 | 6.79 |

**Figure 14.** Substituted quinoxaline sulfonamide derivatives **99–101**.

Ingle and coworkers reported the synthesis of new anticancer quinoxaline sulfonamide derivatives. The 2,3-diphenylquinoxaline derivative **56** was treated with electrophilic compound **47** to produce 2,3-diphenyl quinoxaline-6-sulfonylchloride **57** that further refluxed with primary amines in a basic medium and resulted in the formation of the desired final product, substituted quinoxaline sulfonamides **58** at 53–78% yield (Scheme 10) [136].

**Scheme 10.** Synthesis of quinoxaline sulfonamide derivatives **102**.

These synthesized quinoxaline sulfonamide derivatives were tested for their cytotoxicity in vitro against various cancer cell lines, such as leukemia, non-small cell lung cancer, colon cancer, CNS cancer, melanoma, ovarian cancer, renal cancer, prostate cancer and breast cancer cell lines. Among all the synthesized derivatives, the chlorophenyl-containing moiety quinoxaline sulfonamide **103** and the pyridine moiety-containing quinoxaline sulfonamide **104** (Figure 15) exhibited the greatest anticancer activities against various above mentioned cancer cell lines [136].

**Figure 15.** Substituted quinoxaline sulfonamide derivatives **103–104**.

*2.6. Synthesis of Quinoxaline-Based Benzene Sulfonamide Derivatives with Anti-Inflammatory Activity*

The property of a substance or drug that reduces inflammation or swelling is specified as anti-inflammatory. In other words, nonsteroidal anti-inflammatory drugs (NSAIDs) are drugs that help reduce inflammation, which often helps to relieve pain. NSAIDs can be very effective, and some of the NSAIDS are high-dose aspirin, ibuprofen (Advil, Motrin, Midol) and naproxen (Aleve, Naprosyn), etc.

Ingle et al. synthesized novel quinoxaline sulfonamide derivatives and described their anti-inflammatory activity. The sulfonation of 2,3 diphenylquinoxaline **56** was carried out by the electrophilic agent **47** to produce quinoxaline sulfonyl chloride **57** at 76% yield that was further treated with aliphatic and aromatic substituted amines to obtain the final quinoxaline sulfonamides **105** at 59–85% yield (Scheme 11) [137].

**Scheme 11.** Synthesis of substituted quinoxaline sulfonamide derivative **105**.

The synthesized quinaxoline sulfonamide derivatives were tested for their in vivo anti-inflammatory activity in rat paw edema. The compounds **106–108** (Figure 16) exhibited anti-inflammatory activity, showing inhibition of edema in the range 1.17–4.04% and a mean paw volume of 0.96–0.98 mL. Diclofenac sodium used as reference drug showed, instead, 15.15% inhibition of edema after 30 min with a mean paw volume of 0.85 mL (Table 10) [137].

**Figure 16.** Substituted quinaxoline sulfonamide derivatives **106–108**.

**Table 10.** Anti-inflammatory activity of quinoxaline-based substituted benzene sulfonamide derivatives **104–106**.

| | Anti-Inflammatory Activity | |
|---|---|---|
| Compound | Mean Paw Volume (mL) $\pm$ SEM | % Inhibition of Edema |
| 106 | 0.96 $\pm$ 0.058 | 4.04 |
| 107 | 0.96 $\pm$ 0.052 | 3.82 |
| 108 | 0.98 $\pm$ 0.021 | 1.17 |
| Diclofenac sodium | 0.85 $\pm$ 0.0085 | 15.15 |

## 3. Conclusions

The present review showed the escalating interest of organic and medicinal chemists in the synthesis of quinoxaline scaffolds bearing a sulfonamide moiety to target a variety of diseases. The literature survey cited in this article highlighted and revealed that quinoxaline sulfonamide derivatives show a wide array of biological activities, such as antimicrobial, anti-convulsant, anti-inflammatory, anti-leishmania, anti-tumor and anticancer. Ecofriendly and economical procedures for the synthesis of quinoxaline sulfonamide derivatives were also reported. The structure-based activity findings will be helpful for further modifications on the quinoxaline sulfonamide derivatives. In order to develop future potent therapeutic agents.

**Author Contributions:** Conceptualization: A.I.; resources: S.H., F.B., H.R., R.Z.; writing—original draft preparation: A.I., S.A.; writing—review and editing: K.K.-M., M.M.; All authors have read and agreed to the published version of the manuscript.

**Funding:** Not applicable.

**Institutional Review Board Statement:** Not applicable.

**Informed Consent Statement:** Not applicable.

**Data Availability Statement:** Data sharing not applicable.

**Conflicts of Interest:** The authors declare no conflict of interest.

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
