# Peer review of "Recent Updates on the Synthesis of Bioactive Quinoxaline-Containing Sulfonamides"

_applsci, doi:10.3390/app11125702_

Round 1
Reviewer 1 Report
The manuscript of Irfan et al. is an overview of quinoxaline sulfonamide moiety widely used for the synthesis of drugs and candidate drugs. The authors discuss some compounds citing many literaturessuggesting that the review will be helpful for future drug discovery and development of synthetic pathway. The manuscript may be accepted for publication in this journal after major revision.
The manuscript is confusing and in need of a global reorganization. A preliminary discussion on chemistry, than biological activity and finally the SAR analysis should be performed. The authors should make considerations on SAR and the effects of the substituents on the biological activity. In such form is a mere puzzle of data collected from other manuscripts and the readers cannot have any insight about what the authors declare in the abstract and conclusion “…the review will be helpful for future drug discovery and development of synthetic pathway”.
Several synthetic pathways describe the same reactions. To cite a few: Intermediate 57 is the same compound synthesised in scheme 6, 3 and 10; again, derivatives 71 and 72 are synthesised in schemes 7 and 5 The authors are encouraged to revise all the synthetic pathway and describe all the synthetic approach in only one paragraph, adding considerations about the yields, conditions, technologies used, time of reactions and so on.
Analogously the biological activity should be described through a SAR analys. Many compounds differ to each other only for the position of a substituent (i.e.59 and 102 one is antibacterial and the other one is anticancer. ) The authors should discuss more critically the biological activity and should perform a SAR analysis, not a list of compounds
The English should be attentively revised from a native English
Author Response
Thank you for reviewing our manuscript. The answer to your comments is in the attached file. Sincerely, Mariusz Mojzych

Reviewer 2 Report
The manuscript entitled: “Recent updates on the synthesis of bioactive quinoxaline sulfonamides”
by Ali Irfan, Sajjad Ahmad, Sadam Husain, Haseeba Riaz, Rehman Zafar, Katarzyna Kotwica-Mojzych, and Mariusz Mojzych
proposes a review on the synthesis and biological activity of quinoxaline sulfonamide derivatives.
The manuscript is well structured and may represent a useful tools for scientist expert in this field.
The authors must check and correct some typos present in the manuscritpt (e.g. approach in scheme 3).
Author Response
Thank you for reviewing our manuscript.
Scheme 3 was corrected as revised in the manuscript.
Sincerely,
Mariusz Mojzych
Reviewer 3 Report
Referee report
The sulfonamides are important class of biologically active compounds having the wide application in medical treatment of infections evoked by microorganisms. Their application in other therapies is also well documented. The aim of submitted paper, quinoxaline sulfonamide is located in main stream of medicinal chemistry, but presentation of available data is insufficient. First of all the collected data must be classified into quinoxaline derivatives, where the sulfonamide group is attached to the core aromatic system, in this case heterocycle, by sulfur or (optionally) nitrogen atom and other compounds where heterocycle is a substituent. Separately should be discuss compounds with reduced quinoxaline ring (1,2,3,4-tetrahydroquinoxaline), quinoxaline-2,3(1H,4H)-dione and quinoxaline 1,4-dioxide. In submitted paper it is mixed, the title sulfonamides are discuss together with compounds where the sulfonamide moiety is in peripheral positions.
Details:
Page 2 Line 55 is: Quinoxaline, also known as benzopyrazine, was formed by the combination of pyrazine 2 and benzene 3 rings.
This is not precisely defined benzene is fused with carbons 2 and 3 of pyrazine ring. Table 1 is superfluous, these data can added to the text.
In figure 5 compounds 17, 25, 26, 28 and 33 are not quinoxaline derivatives, they contain quinoxaline moiety as a substituent. 27 and 32 are derivatives of 1,2,3,4-tetrahydroquinoxaline.
Compounds described in chapter 2.1 are not quinoxaline but 2-aminobenzo[d]thiazole-6-sulfonamide derivatives, where quinoxaline is a substituent. Similarly chapter 2.2., presented compounds belong to benzenosulfonamides.
Page 13 line 229 micrograms have international abbreviation µg.
Page 14 line245 type writing error quinaxoline please change to quinoxaline
Page 15 line 260 - 269 the activity data please collect in table
In scheme 5, Figure 11 and scheme 7 are depicted derivatives of quinoxaline-2,3(1H,4H)-dione
Page 19 line 329 is V. cholorie, it means Vibrio cholera ?
Page 20 line 338 In table 6 please indicate Gram positive and Gram negative bacteria.
Page 22 line 379 scheme presents synthesis of quinoxaline 1,4-dioxide derivatives. Similarly Fig. 13
Page 24 line 420 Scheme 9 and Fig. 14 presents classical benzenesulfonamide derivatives with peripheral quinoxaline substituent.
Page 25 table 9, Is known reference compound?
Page 26 line 447 – 449 please add the numbers for particular cell lines e.g. HTC-116 (colon cancer) etc.
Author Response

(The authors gave the same response as above.)

Reviewer 4 Report
Dear Authors,
I had the privilege to review your manuscript and I am impressed by the extensive review of quinoxaline derivatives and their many applications.
To make your work more comprehensive and resourceful for the esteem readers, I suggest correcting some grammatical & mechanical errors I highlighted in the manuscript.
I also noticed that some of your references didn't follow any particular format. In the references, I highlighted those that need to be rewritten. I suggest that authors stick to one type of citation format throughout their references.
Looking forward to receiving your updated manuscript.
Regards,
Reviewer

Author Response

(The authors gave the same response as above.)

Round 2
Reviewer 1 Report
I would like to thank the authors to reply to my comments highlithing the paragraphs reported in the first version of the manuscript in the reviewer's reply.
"The intermediates of different reactions are same because same nucleus quinoxaline is synthesized with same conditions or with different conditions but the end products are different or have different substitution pattern in final products."
Right! and this is why I suggest to discuss the chemistry alone and the biological activity in a different section.
The authors conclude the manuscript declaring that this review aims to be helpful in drug discovery and development. My question is still the same. How?
Author Response
Dear Reviewer,
Thank you very much for all your sugestions.
According to your last opinnien the last sentence in our manuscript: "This review article has the aim to be helpful in future drug discovery and development." we have replaced by the new one: "This review article has the aim to summarize the recent approach to the preparation of sulfonamides bearing quinoxaline derivatives and their various biological activity."
I hope that in this form our manuscript looks better and meets your expectations.
Best regards,
Mariusz Mojzych
Reviewer 3 Report
Thank you for accepting suggestions formulated to first version of manuscript. New title of paper is much capacious and covers all discussed derivatives. It cancels my objections.
Author Response
Dear Reviewer,
Thank you very much for all your sugestions.
Best regards,
Mariusz Mojzych
Round 3
Reviewer 1 Report
I thank the authors for changing made in the conclusion section.
I would like to see the same correction also in the last sentence of the abstract
Author Response
Dear Reviewer,
Thank you for all your comments. They are very valuable to us and have improved the quality of our manuscript. The abstract has been modified according to your suggestion.
Best regards.
Mariusz